# MEMORY-EFFICIENT ALGORITHM DISTILLATION FOR IN-CONTEXT REINFORCEMENT LEARNING

## ABSTRACT

It's recently reported that by employing the superior In-context Learning (ICL) ability of autoregressive Transformer, a method named *Algorithm Distillation* (AD) could distill the whole Reinforcement Learning process into neural network then generalize to *unseen* scenarios with performance comparable to the distilled algorithm. However, to enable ICL, it's vital for self-attention module to have a context that spans cross-episodes histories and contains thousands of tokens. Such a long-range context and the quadratic memory complexity of self-attention pose difficulty on applying AD into many common RL tasks. On the other hand, designing memory efficient Transformers for *long-range document modeling* is itself a fast-developing and fruitful field, which leads to a natural question: *Could Efficient Transformers exhibit similar in-context learning ability and be used for Memory-Efficient Algorithm Distillation?* In this paper, we firstly build a benchmark suite that is thorough, efficient and flexible. Thanks to it, we perform extensive experiments and verify an existing method named *ERNIE-Docs* (ED) could offer competitive performance with significantly reduced memory footprint. With systematic ablation studies, we further investigate various facets influencing the ICL ability of ED and provide our own insights into its hyperparameter tuning.

## 1 INTRODUCTION

Building decision-making agents that could generalize to various environments and tasks has been a long-standing goal of Reinforcement Learning (RL). Recently, following the trends in Computer Vision (CV) (Dosovitskiy et al., 2021) and Natural Language Processing (NLP) (Brown et al., 2020; Devlin et al., 2019), multiple works have explored on pretraining deep autoregressive Transformers on large-size behavior dataset to obtain agents with strong generalization and adaptation ability (Reed et al., 2022; Lee et al., 2022; Octo Model Team et al., 2024). Among them, *Algorithm Distillation* (AD) takes an unique approach and proposes that by employing the superior In-Context Learning (ICL) ability of autoregressive Transformer, the whole RL process could be distilled into a neural network then be 'replayed' on new, possibly unseen scenarios.(Laskin et al., 2023)[1]

Analogous to causal Language Modeling (LM), AD requires an autoregressive Transformer with context spanning cross-episodes interaction histories to perform *credit assignment*, *exploration-exploitation balance* and *policy improvement* that are vital to a RL algorithm. Differently and crucially, all of these abilities are implicitly modeled in the weights of Transformer network instead of being designed by human experts and implemented as hard-coded programming languages as in Haarnoja et al. (2018); Fujimoto et al. (2018); Schulman et al. (2017).

However, this cross-episodes context encompasses thousands of tokens, resembling **long-range document modeling** in NLP: In typical RL settings, an episode contains hundreds of transitions each comprising 4 *components* (*state*, *action*, *reward* and *terminal*):

$$2 \sim 4 \text{ (cross-episode context length is needed)} \times 500 \text{ (transitions in an episode)}$$
$$\times 4 \text{ (components in a transition)} = 4k \sim 8k \text{ (tokens)}$$

Combined with the quadratic memory cost of self attention, this requires excessively large memory footprint (around 40-160GB under batch size 10, embedding size 128 and half-precision) and prevents RL researchers from readily reproducing and improving algorithms in AD's setting.

---

[1]We use term 'AD' to denote both the problem setting and vanilla Transformer used in the original paper.

On the other hand, designing memory & computation Efficient Transformer (ET) for long-range document modeling is itself an important and developing research field in NLP where many well-established works have emerged (Dai et al., 2019; Ding et al., 2021; Tay et al., 2022). Therefore, a natural question arises: *Could Efficient Transformers exhibit similar in-context learning ability and be used for Memory-Efficient Algorithm Distillation?* If so, it could combine the best of both worlds and significantly boost future researches in this line.

However, to the best of our knowledge, there is still no works investigating this question. The reasons are multi-fold: 1) It still lacks a consensus on how to design benchmarks to thoroughly test the ICL ability of a given algorithm. 2) As of a meta-RL setting, the implementation of above benchmark shall be computation-efficient for parallel data collection and evaluation.

Therefore, in this work, we firstly design an efficient and flexible benchmark suite to thoroughly test the ICL performance of a given method. After this, with comprehensive experiments, we discover *ERNIE-Docs* (ED) (Ding et al., 2021), an improved variant of *Transformer-XL* (Dai et al., 2019), could obtain competitive performance with a significantly reduced memory requirement. Finally, with systematic ablation studies, we examine various factors influencing the ICL performance of ED and provide our own insights in its hyperparameter tuning. In summary, our contributions are:

- We provide a benchmark suite covering the whole AD process from data collection to meta training & evaluation. With 8 representative settings, we could thoroughly test a method's ICL ability in cases of sparse reward, credit assignment, dense information and exploration-exploitation balance.
- We employ JAX (Bradbury et al., 2018) for computation efficiency and achieve parallel execution on *10k* different environments. Our code is also compatible with Pytorch (Paszke et al., 2019) for its broader Transformer-related ecosystem.
- Based on our benchmark, we find **ERNIE-Docs** obtains competitive performance with a significantly reduced memory requirement. We also perform systematic ablation studies to demonstrate how various factors affect its performance and provide our own insights.

## 2 RELATED WORK

**Sequence Modeling in Reinforcement Learning**. Interpreting RL as a problem of sequence modeling and employing Transformer for it has been a popular research direction since the recent success of Transformer in NLP field (Radford et al.; Devlin et al., 2019). Chen et al. (2021) and Janner et al. (2021) firstly introduced Transformer into model-free and model-based RL respectively and achieved promising results in many tasks. As a pioneering work, Laskin et al. (2023) firstly reported that by using autoregressive Transformer, in-context RL could be achieved with generalization to unseen scenarios. Following their work, Dai et al. (2023); Zisman et al. (2024); Sinii et al. (2024) investigated various facets to improve AD, such as training on noisy dataset and generalizing to unseen action space. Differently, our work targets on the memory cost of vanilla AD.

**Meta Reinforcement Learning**. Meta RL aims at empowering RL algorithms the ability to quickly adapt to new environments and tasks with limited amount of *on-site* samples. Generally, it could be classified into 2 branches (Beck et al., 2023): 1) in-weight meta RL where methods like MAML (Finn et al., 2017), ProMP (Rothfuss et al., 2022) perform gradient descend on parameters of neural network for fast adaptation. 2) in-context meta RL where adaptation emerges as context of environment interactions gets populated or updated as in methods like RMA (Kumar et al., 2021) and Prompt-DT (Xu et al., 2022). Different from these works whose goal is fast adaptation of episode return, AD aims to distill existing RL algorithms then generalize to new scenarios.

**Transformer for Long-range Tasks**. Designing Efficient Transformers to increase the performance on long-range tasks has been a very important field (Burtsev et al., 2021; Tay et al., 2022). Dai et al. (2019) proposed integrating recurrence with Transformer's attention mechanism and achieved superior performance on long-range tasks. Later work like Rae et al. (2019) also studied to 'compress' past information into recurrence module and further improved Transformer's performance over tasks on long-range document modeling while not drastically increasing context length. Besides, methods like (Gu & Dao, 2023; Wang et al., 2020) tried to design new attention mechanisms with only linear complexity to context length. Since this field is both fruitful and fast-developing, in this work we choose to explore how these promising methods could be used in AD's setting such that RL researchers could be freed to explore on ideas more related to RL's problems.

# 3 PROBLEM FORMULATION

## 3.1 PRELIMINARY

**Algorithm Distillation**. Consider a meta-RL setting, where a MDP $\mathcal{M}_i$ is sampled from a given distribution. For each sampled MDP, we use a RL algorithm $\mathcal{A}$ (e.g. Q-Learning (Sutton & Barto, 2018) for discrete setting or SAC (Haarnoja et al., 2018) for continuous setting) to search for the optimal policy $\pi$ whose performance is measured by *episode return* $\sum_{i=0}^{h} R_i(s, a)$ where $h$ is the maximum episode length and $R_i$ is the reward for $i$th transition. Then, by collecting the whole history $\tau_i$, we get a dataset $D = \{\tau_i = (s_0, a_0, r_0, d_0, ..., s_T, a_T, r_T, d_T)\}_{i=1}^{n}$ where $s$ is *state*, $a$ is *action*, $r$ is *reward*, $d$ is *terminal* and $\tau_i$ is the learning history of $\mathcal{A}$ on $\mathcal{M}_i$. Note $\tau$ may contain many episodes ($T \gg h$) in which the performance of $\pi$ gradually improves.

AD proposes by performing autoregressive training, $\mathcal{A}$ could be distilled into a neural network $\phi$:

$$\mathcal{J} = \arg\max_{\phi} \mathbb{E}_{\tau \sim D; s,a,r,d \sim \tau} \left[ \phi(a_t | s_{0:t}, a_{0:t-1}, r_{0:t-1}, d_{0:t-1}) \right] \tag{1}$$

Then the weight-frozen $\phi$ could be used on unseen $\mathcal{M}_j$ and yield similar learning process as $\mathcal{A}$.

**Self Attention**. For a token sequence $x \in R^{s \times d}$ of length $s$, self attention firstly transforms $x$ into three matrices: *query* $Q = xW^Q$, *key* $K = xW^K$ and *value* $V = xW^V$, where $W^Q, W^K, W^V \in R^{d \times d}$. Then *scaled dot attention* is applied between any 2 elements in $x$:

$$\text{Self-Attention}(Q, K, V) = \text{Softmax}(\frac{QK^T}{\sqrt{d}})V \tag{2}$$

**Recurrence Transformer**. Transformer-XL (XL) (Dai et al., 2019) proposes to integrate recurrence into attention to obtain longer context length. As shown in Figure 1, the token sequence is split into equal-length chunks and the Transformer processes these chunks sequentially from beginning to ending. During this, the hidden state of a previous chunk is preserved and acts as additional memory (*key* and *value*) when processing the following chunk. For the $i$th transformer layer $L^i$, the input and output are:

$$h^i_{kc:(k+1)c} = L^i(q = h^{i-1}_{kc:(k+1)c}, k = v = [\text{sg}(h^{i-1}_{(k-1)c:kc}), h^{i-1}_{kc:(k+1)c}]) \tag{3}$$

where $h^i_k$ represents hidden embedding output by $i$th layer for $k$th chunk, $c$ is chunk length, sg is *stop-gradient*, $q$,$k$,$v$ are *query*, *key*, *value*, respectively and $[\cdot, \cdot]$ concatenates 2 vectors. As an improved version of XL, ERNIE-Docs (ED) (Ding et al., 2021) chooses to shift the preserved hidden embeddings one-layer down for attention computation (difference to XL is in red):

$$h^i_{kc:(k+1)c} = L^i(q = h^{i-1}_{kc:(k+1)c}, k = v = [\text{sg}(h^i_{(k-1)c:kc}), h^{i-1}_{kc:(k+1)c}]) \tag{4}$$

Through this simple yet effective modification, ED could have a context that is *theoretically indefinitely long*, as shown in Figure 1, right.

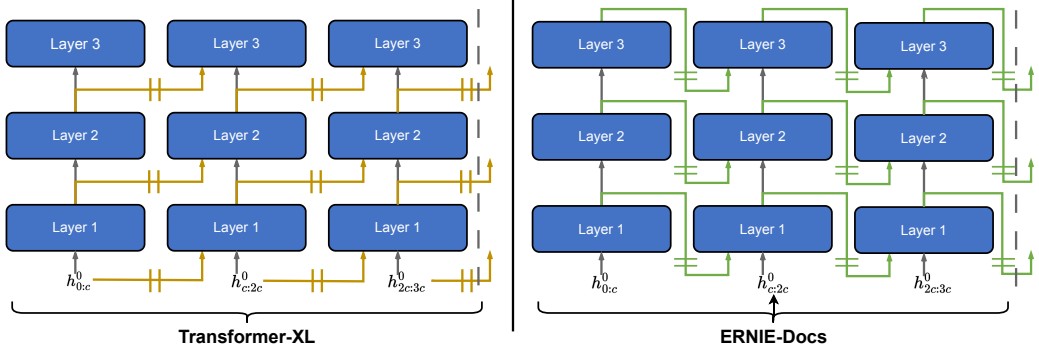

Figure 1: Framework of Transformer-XL and ERNIE-Docs. The double line represents *stop gradient*. The output of last layer is omitted for simplicity

## 3.2 MEMORY COST ANALYSIS

We present a formal analysis of memory cost for vanilla Transformer and Recurrence Transformer (abbreviated as RT and denoting XL and ED). For the sake of simplicity, only attention module is focused as other modules like tokenizer, feedforward and layernorm layer are the same in our settings.

**Vanilla Transformer**. Let's consider an input $x$ of shape $(b, s, d)$ where $b$ is batch size, $s$ is sequence length and $d$ is hidden dimension. For a mutli-head attention layer with $n$ heads. This requires the following amount of memories:

$$
\begin{aligned}
&3(d^2 + d)\,(\text{weights \& biases}) + 3(d^2 + d)\,(\text{gradients of weights \& biases}) \\
&+ 3bsd\,(\text{Q, K, V}) + nbs^2\,(QK^T) + nbs^2\,(\text{attention score}) + bsd\,(\text{final output}) \\
&= 6d^2 + 6d + 4bsd + 2nbs^2 \propto s^2
\end{aligned}
\tag{5}
$$

**Recurrence Transformer**. Let's consider a chunk length of $c$ and a recurrence length of $m$. Then for each step of RT processing, the chunk input $x_c$ is in shape $(b, c, d)$ and recurrence $x_m$ is in shape of $(b, m, d)$. The amount of memories for RT is:

$$
\begin{aligned}
bmd\,(\text{recurrence}) &+ 3(d^2 + d)\,(\text{weights \& biases}) + 3(d^2 + d)\,(\text{gradients of weights \& biases}) \\
&+ bcd\,(\text{Q}) + 2b(c + m)d\,(\text{K, V}) + nbc(c + m)\,(QK^T) \\
&+ nbc(c + m)\,(\text{attention score}) + bcd\,(\text{final output}) \\
&= 3bmd + 6(d^2 + d) + 4bcd + 2nbc(c + m) \propto c(c + m)
\end{aligned}
\tag{6}
$$

From above analysis, the memory cost is proportional to $s^2$ for vanilla Transformer and is proportional to $c(c + m)$ for RT methods. However, in the setting of AD, cross-episode context is needed for vanilla self-attention based Transformer and encompasses thousands of tokens. On the other hand, the recurrence module in RT could serve as an additional memory for important information and reduce the need of contiguous context which is unnecessarily long. That is to say, with $c + m \ll h < s$ where $h$ is episode length, the memory cost of RT could be greatly reduced than vanilla Transformer.

## 4 BENCHMARK DESIGN

It's still an open question over how to design a benchmark suite to test the ICL ability of AD methods. Firstly, the benchmark needs to be **thorough** enough and include representative RL settings. Besides, as an in-context meta RL setting, the benchmark needs to be **computation-efficient** and support parallel environment interactions for both data collection and meta evaluation. Finally, the benchmark shall also be **flexible** and compatible towards the most ecosystems.

In this work, taking all above considerations into account, we design a benchmark suite that is computation-efficient thanks to environment parallelism in JAX and flexible since it also supports building algorithm in pure Pytorch. We also carefully design environments & tasks to thoroughly test given methods' ICL performance. An illustrative framework of our benchmark is shown in Figure 2. The further details could be found in Appendix A.3.

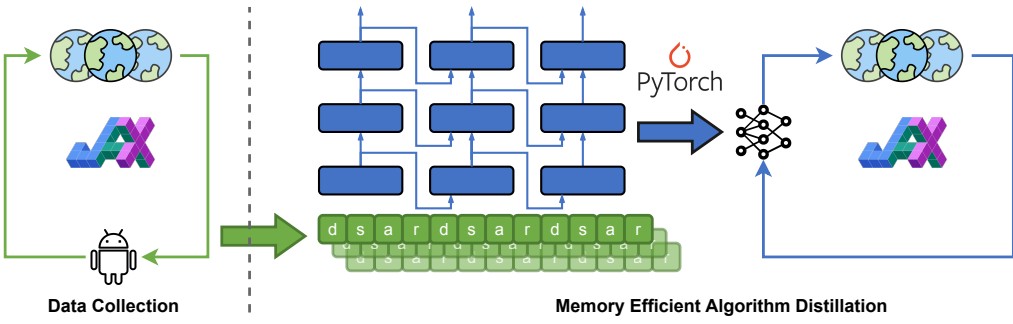

Figure 2: Framework of benchmark process

**Environments**. We choose `GridWorld` where the observation is 2-dimensional coordinates ($x,y$) and action contains 5 options: *up, down, left, right, none*, as shown in Figure 3.

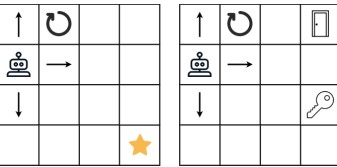

- `DarkRoom` (dr). The grid is 9x9 large. The episode length is 20. The agent spawns in the middle of it with a randomly placed invisible goal. This environment tests the agent's ability to explore in an unknown environment to find the 'goal' then remember its location for exploitation.

Figure 3: Environments

- `DarkKeyToDoor` (dktd). The grid is 9x9 large. The episode length is 50. An agent spawns in the middle of it. A *key* and a *door* are randomly placed and invisible to agents. Note this environment is more challenging compared to `DarkRoom` because it features 2-phase sequential tasks as the 'key' could only be interacted once and the 'door' could only be opened when agent holds the key.
- `DarkRoomLarge` (dr**l**). A larger verion of `DarkRoom`. The grid size is 13x13. The episode length is 50. Note the increase in grid size generally results in exponentially increased difficulty in exploration and memorization.
- `DarkKeyToDoorLarge` (dktd**l**). A larger version of `DarkKeyToDoor`. The grid size is 11x11. The episode length is 70.

**Tasks**. 3 representative tasks are designed: *normal*, *dense* and *quick* to test the algorithm's RL ability like credit assignment and sparse reward, as shown in Figure 4.

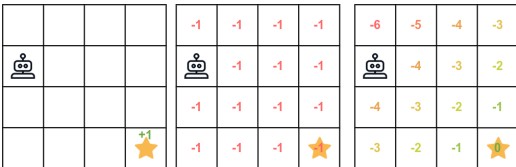

- `normal` (**n**). In `DarkRoom`, if the agent finds the goal, a reward of +1 is granted and the episode terminates. For `DarkKeyToDoor`, if the agent finds the key, a reward of +1 is granted *only once*. Then if the agent finds the door with key on hand, another +1 reward is granted and the episode terminates. For other cases, the reward is 0. This is a sparse reward

Figure 4: Illustration of tasks

setting and also tests the memorization ability of agent as it needs to remember the location of 'target' ('goal' for `DarkRoom` or 'key'/'door' for `DarkKeyToDoor`) and discover that the 'key' only grants one-time reward.
- `dense` (**d**). For both `DarkRoom` and `DarkKeyToDoor`, the reward for each step is the negative $L_1$ distance from agent to current 'target'. The episode still terminates once the goal is found or door is opened. For other cases, reward is 0. This setting features dense reward and is often overlooked in previous works.
- `quick` (**q**). For all steps, the reward is -1. Only when the agent finishes the task, the episode would terminate which encourages agent to finish the episode as soon as possible. This is the most challenging setting and when used with `DarkKeyToDoor`, a strong ability of credit assignment is required since agent gets no immediate signal when finding the 'key'.

In total, we select 8 *settings* from above environments and tasks : dr, dr**ln**, dr**lq**, dr**ld**, dktd, dktd**ln**, dktd**lq**, dktd**ld** with the format {env. name}{task variant}. Note for certain setting, the gridsize and episode length may be slightly modified, please refer to Appendix A.3 for details.

**Algorithms**. We choose the following algorithms for testing on above setting. Their details are in Appendix A.4:

- `SOURCE`. This is the original algorithm used to collect the dataset. We use Q-Learning for all tasks. Notably, for `DarkKeyToDoor`, the state of `SOURCE` is augmented with one extra dimension: *has key or not* for *Markovian property* while in dataset and the following training/evaluation of AD methods, this extra dimension is removed. In our settings, the learning curve of `SOURCE` is regarded as *ORACLE* and shall be mimicked by other methods.
- `ADF`. This is the original AD algorithm implemented manually by us. The F means we use long context length which is often ~2.5 times of episode length following recommendations in the original paper (Laskin et al., 2023).
- `ADR`. This is the AD method with a reduced context length for a fair comparison to other memory-efficient algorithms. In most cases, the context length of `ADR` is less than a half of episode length. For details, please refer to Appendix A.4.4.

- `MEM`. This is the method introduced in Burtsev et al. (2021). Specifically, we use `MemCtrl` where a set of learnable embeddings acts as global memories, gets prepended to all sequence before attention layers and discarded after its processing.
- `XL`. This is the Transformer-XL method as described above and in Dai et al. (2019). Notably, this method (and the following `ED`) requires Truncated-Backprop-Through-Time (TBPTT) training.
- `ED`. An improved version of `XL` with *theoretically indefinitely long* context length.

In the following contents, we use Efficient Transformer (ET) to denote methods of `MEM`, `XL` and `ED` for simplicity.

**Training & Evaluation**. As this is a setting of in-context meta RL, for each of 8 settings defined above, we firstly collect a dataset containing training process of `SOURCE` on 10k environments with various locations of 'target'. Then, we train each above algorithm on the dataset and evaluate on 100 new environments whose 'target' locations are disjoint against training ones, i.e. out-of-distribution (OOD) evaluation. For comparing the performance of each algorithm, we plot its curve of trajectory return against ICL process[2], and the more its curve mimics the curve of `SOURCE`, the better performance it exhibits.

**Extensibility**. Thanks to the flexibility and broad ecosystem of JAX and Pytorch, our benchmark could be easily extended to support more environments, tasks & algorithms. For example, via MujoCo's MJX (Todorov et al., 2012), environment parallelism with domain randomization could be easily enabled for continuous settings like `Walker`, `HalfCheetah` and `Humanoid`. Plus, it's also compatible with Pytorch with which many existing Efficient Transformers are implemented. Hence, we plan to publish this benchmark and would continue to add representative environments, tasks & algorithms into it for more comprehensive results.

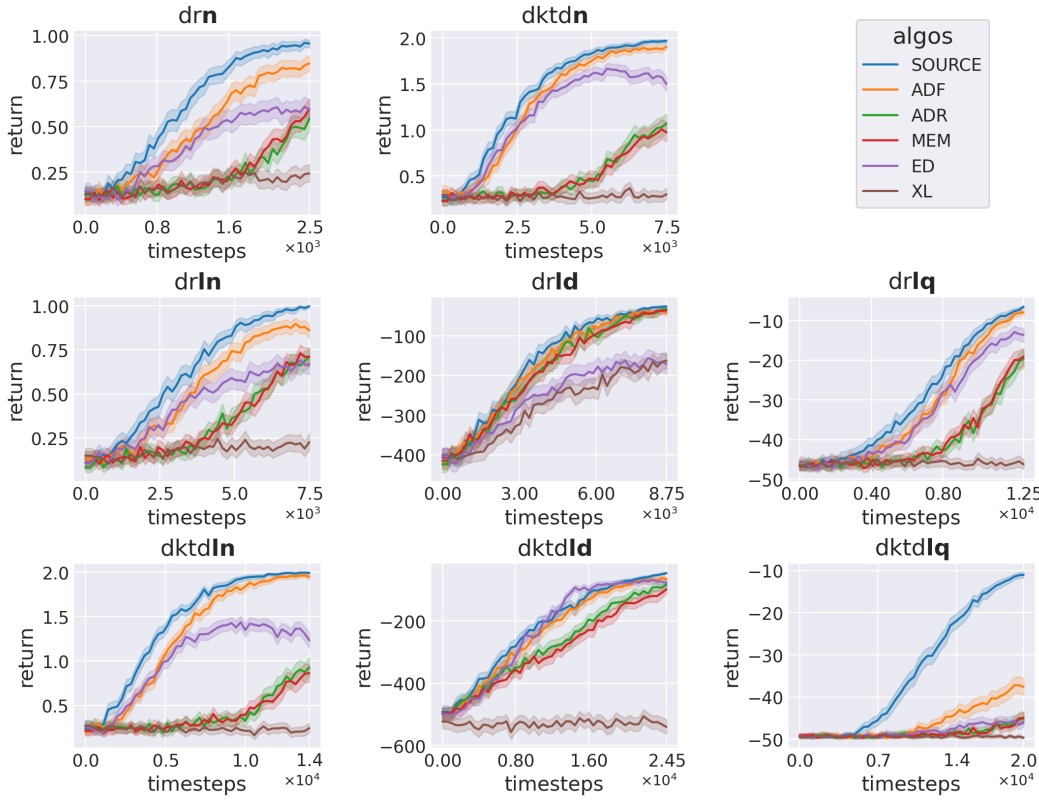

Figure 5: Results of main experiments. The shaded area is 95% confidence interval

---

[2]The ICL process here is indeed the in-context RL process which spans many episodes with gradually improved episode returns.

# 5 EXPERIMENTS

## 5.1 MAIN RESULTS

We present our main experiment results of training above 6 algorithms on 8 settings. The learning curves are shown in Figure 5 and the memory consumptions of these methods are present in Table 1.

Table 1: Memory consumption of various methods

| Settings | Episode Length | ADF [α] | ADR [α] | MEM [β] | XL [β] | ED [β] |
|---|---|---|---|---|---|---|
| dr | 20 | 50 \| 12.75 GB [γ] | 10 \| 6.42 GB | 10 \| 10 \| 6.93 GB | | |
| dktd | 50 | | | | | |
| dr**ln** | | | | | | |
| dr**lq** | 50 | | | | | |
| dr**ld** | | 125 \| 37.33 GB | 25 \| 10.65 GB | 25 \| 50 \| 13.96 GB | | |
| dktd**ln** | 70 | | | | | |
| dktd**lq** | 50 | | | | | |
| dktd**ld** | 70 | | | | | |

$\alpha$ For AD methods, the format is {context length} | {memory consumption}.
$\beta$ For ET methods, the format is {chunk length} | {recurrence capacity} | {memory consumption}.
$\gamma$ This is the total GPU memory consumption measured on device.

From the results, several interesting observations could be drawn:

**Dense information helps Algorithm Distillation**. Notably, for dr**ld** and dktd**ld**, most algorithms obtain quite decent ICL performance, even for MEM and ADR. We conjecture this is due to the rich guidance provided by the reward signal in *dense* task hence a sub-episode context could sufficiently prompt algorithm for the correct behavior.

**Tasks requiring credit assignment are challenging**. For dktd**lq**, almost all algorithms fail, showing this is a quite challenging setting since the acquisition of 'key' provides no immediate signal to agent. Even for ADF, the performance is inferior compared to SOURCE, indicating a context length of more than 2.5 episode length may be needed. How to increase the ICL performance of AD methods and reduce their memory cost on this setting would certainly be a very important and promising directions for RL researchers.

**Context Length is critical for vanilla AD method**. From above results, we find compared to ADF, ADR often obtains significantly worse performance, except on dense-information tasks. This aligns with our expectation that the self-attention context in AD needs to span multiple episodes to enable a decent AD performance. However, combined with quadratic memory cost of self-attention, this long-range context causes difficulty for researches to experiment with, as also shown in Table 1 where the ADF needs significantly more GPU memory than other methods.

**ED for Memory-Efficient Algorithm Distillation**. For ET methods, we find:

- MEM obtains quite similar performance compared to ADR, indicating its global memory doesn't help in algorithm distillation.
- XL, with just slight differences to ED, also obtains insufficient performance, even compared to ADR. Even after architecture tuning, we still find the performance of XL is unstable and argue this results from its limited ability to utilize information in recurrence.
- ED, on most of settings could obtain a competitive performance compared to ADF and SOURCE thanks to its clever usage of recurrence memory. On the other hand, there is still room for further improvements as its learning curve tends to slightly drop near the ending phase, suggesting it struggles to perform exploitation compared to SOURCE and ADF. With some hyperparameter tuning and architecture tweaking, this drop could be mitigated in some extend, as shown in Section 5.2.4. Nevertheless, further investigation and mitigation to this phenomenon would still be an interesting and valuable future work.

## 5.2 ABLATION STUDIES

To better illustrate how various facets affect Memory-Efficient Algorithm Distillation, we design several ablation studies testing the positional encoding, Transformer model size, attention module and context length & memory capacity.

### 5.2.1 POSITIONAL ENCODING

Positional Encodings (PE) like *sinusoidal encoding* (Vaswani et al., 2017), *learnable embedding* and *RoPE* (Su et al., 2023) are critical components for attention-based Transformer models and have been widely studied in NLP.

Interestingly, in our experiments, we find the usage of *global* positional encoding has unique advantages over local-context-based ones common in NLP. Without change in embedding mechanism, the 'global' PE assigns to all the tokens in the whole learning process a positional embedding while context-based PE only considers tokens in the context window. We argue global PE is more suitable for in-context RL as it's intuitive for RL algorithms to take different strategies in different learning phase, such as exploration in the beginning and exploitation in the ending.

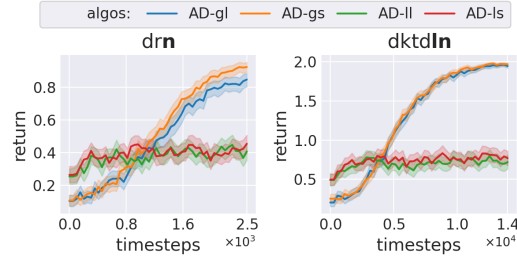

Figure 6: Performance when using different PEs

Besides, we also design 2 experiments to verify this: 1) **Global PE helps training**. As shown in Figure 6, we test 4 PEs on standard `ADF` method. The 4 abbreviations represent *global-learnable* (gl), *global-sinusoidal* (gs), *local-learnable* (ll) and *local-sinusoidal* (ls), respectively. From the results, global positional encoding significantly helps stabilize and ease the training of Algorithm Distillation. 2) Additionally, **global PE could tune the behavior of distilled algorithm**.

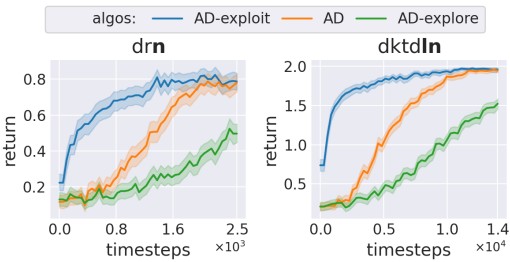

Figure 7: Global PE controls algorithm behavior

By tweaking the positions passed to the PE module, we could tune the algorithm to be more *exploratory* or more *exploitative* without changing the input sequence. As shown in Figure 7, when we tweak the positions passed to Transformer model to be larger, the performance of AD tends to increase since it focuses more on *exploitation*. While if we tweak the position to be smaller, the AD becomes more *exploratory*. Therefore, we choose *global learnable* PE in main results and the followings.

> **Takeaway**
>
> Global Positional Encoding helps training and controls exploration-exploitation behavior.

### 5.2.2 TRANSFORMER MODEL SIZE

Previous works in NLP and AD have found that with the increase of model size, Transformer models exhibit better in-context learning ability. In this section, we also study whether this phenomenon happens on `ED` for memory-efficient AD. Specifically, for dr**n** and dkld**ln**, we test the effects of reducing the model size [3] over the ICL ability of `ED`. The results are shown in Figure 8, where the format of the legend is {algo name}-{number of attention head}-{token dim.}-{feedforward dim.} and 'edo' means the original `ED` used in the main experiment which has the largest model size.

From the results, we could find with the model size increasing, the ICL performance of `ED` is also increased. Even for dr**n** which is a rather small environment, a model with 64 attention heads, 512

---

[3]the number of attention heads are also reduced to maintain a similar attention head dimension.

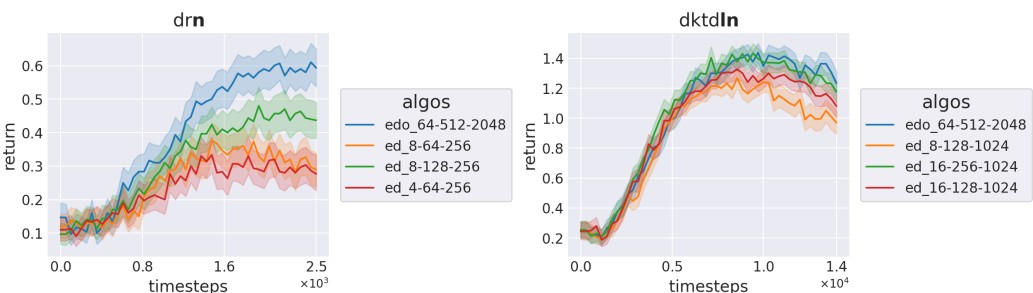

Figure 8: Performance of large model vs small model

token dimension and 2048 feedforward dimension is needed for a decent performance. On dktd**ln**, the effect of increasing model size also exists but is in a less significant extend.

> **Takeaway**
>
> Larger model enables better performance for ED-based Memory-Efficient AD.

### 5.2.3 RECURRENCE-BASED ATTENTION

Attention mechanism is key to the success of Transformer models. In this work, as a recurrence-based Transformer method, ED also utilizes attention to extract important information from recurrence memory. Therefore, we also test how various attention settings influence its ICL performance in the following experiments.

**Does ED attend to the recurrence memory?** It would be natural to expect the outstanding ICL performance of ED results from the usage of its recurrence memory. Therefore, we visualize the attention ratio of ED for all 8 settings. In details, we sum the softmaxed attention scores for tokens in ED's recurrence memory: 1 means ED only attends to

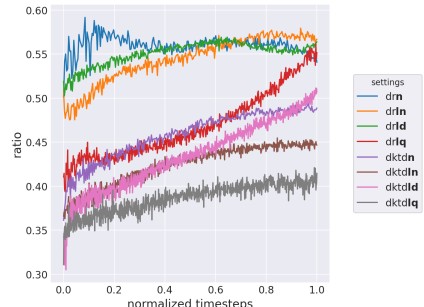

Figure 9: Recurrence attention ratio

recurrence memory and ignores chunk tokens while 0 represents vice versa, then average this *ratio* over all attention heads and all 100 environments in evaluation set and plot it against the learning progress, as shown in Figure 9.

From the results, it clearly shows ED attends to its recurrence memory as all the attention ratios are between 0.3∼0.6. Besides, several interesting observations could be drawn from it: 1) For dktd**lq** where ED obtains no performance due to the challenge of credit assignment, its attention ratio is also the lowest, indicating ED fails to extract useful information from the recurrence memory. 2) For dktd**ld** and dr**lq** where ED obtains near-optimal performance, the attention ratio shows a notable increase near the ending of progress. While for the other settings where ED's performance declines at the ending of process, their attention ratios doesn't show such increase.

**How does the number of attention head influences ICL performance?** In this section, we test the effects of number of attention heads of ED over its ICL ability. Specifically, we change the attention head number from 64 to 32 and 128 respectively. Note we don't change the embedding dimension hence this would result in changes in dimension of attention head. The results are shown in Figure 10 where the format of legend is {algo name}_{amount of attention head} and 'edo' is the original ED used in main experiment with 64 heads.

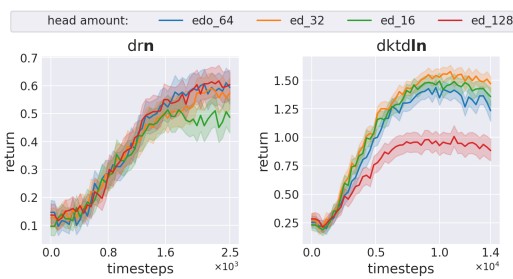

Figure 10: Effects of amounts of attention heads

From the results, we could find in dr**n**, a relatively simpler setting, the amount of attention heads doesn't have a significant influence on `ED`'s ICL performance. While in dktd**ln**, increasing the amount of attention head (which also reduces the dimension of attention head) has a negative impact over the ICL ability. On the other hand, slightly decreasing it to 32 could increase performance slightly. Note this doesn't conflict with the results presented in Section 5.2.2 as we change both the number of attention head and its dimension there. Taking these together, we recommend setting attention head dimension to be 8 or 16 with at least 32 or 64 attention heads as a good start.

> **Takeaway**
>
> Use 32 or 64 attention heads with dimension of 8 or 16 for a good start.

### 5.2.4 CHUNK LENGTH & RECURRENCE CAPACITY

In this section, we study the effects of chunk length & recurrence capacity over the performance of `ED`'s ICL performance. Specifically, we designed 5 settings: 1-1, 1-2, 1-3, 2-1, 2-2, 2-3 where the first digit represents the relative ratio on chunk length while the latter digit represents the relative ratio on recurrence capacity. The results are shown in Figure 11 where the format of legend is `{algo name}_{chunk length}-{recurrence capacity}` and 'edo' is the original setting.

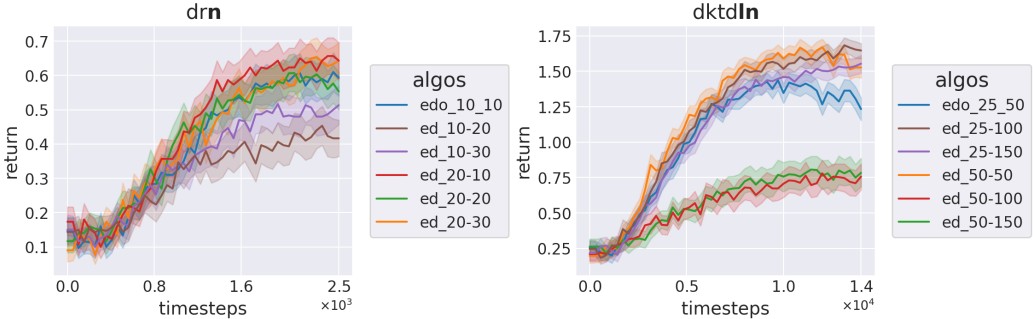

Figure 11: Effects of chunk length & recurrence capacity

The experiments show intricate results: 1) Increasing recurrence capacity alone may not result in better performance as in dr**n**, `ed_10-20` and `ed_10-30` obtain worse performance than `edo_10_10`, and `ed_20-20` and `ed_20-30` also show decreased performance compared to `ed_20-10`. 2) For dktd**ln**, however, several settings obtain boosted performance compared to `edo_25-50`: `ed_50-50`, `ed_25-100` and `ed_25-150` while `ed_50-50` and `ed_50-100` obtain significantly worse performance. In summary, we recommend setting them to be 1:1 with careful tuning of values.

> **Takeaway**
>
> Keep balanced ratio between chunk length and recurrence capacity and tune the value.

## 6 CONCLUSION

In this work, we study the problem of reducing the excessive memory requirement of Algorithm Distillation (AD) via existing Efficient Transformers (ET). With our efficient, flexible and versatile benchmark, we discover the ERNIE-Docs (ED) could serve as a simple yet effective method for Memory-Efficient Algorithm Distillation and illustrate how various factors like positional encoding, model size, attention module and context length & memory capacity influence its performance.

Besides, there is still room to further improve ED's performance and better interpret/visualize its internal attention mechanism which we choose to leave as future works. Also, because ET is an fast-developing field and limited by computation resources, we cannot test all promising ET methods in this work. Nevertheless, we will release our benchmark code for further researches in the line of Memory-Efficient AD.

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

# A APPENDIX

## A.1 PSEUDO-CODE FOR AD'S TRAINING & EVALUATION PROTOCOL (NEWLY ADDED)

We list the training & evaluation protocol of AD-like algorithms used in this work in Algorithm 1.

---

**Algorithm 1:** Training & Evalutaion Protocol of AD-like algorithms

---

**Input:** Dataset $D_\tau$ obtained by running RL algorithm $\mathcal{A}$ on $k$ different environments for $T$ timesteps which span multiple episode length $h$ (typical setting: $k = 10^4$, $T = (100 \sim 1000)h$)

**Output:** a transformer-based nerual network $\phi_\mathcal{A}$ showing comparable learning process of $\mathcal{A}$ on new scenarios

1 $\phi_\mathcal{A} \leftarrow$ random init.
2 **while** *training not converged* **do**
3     $\tau \leftarrow$ sample_chunk$(D_\tau)$
4     $\phi \leftarrow$ SGD$(\phi_\mathcal{A}, \tau)$                 // autoregressive training
5     **if** *time to evluate* **then**
6        $rewards \leftarrow$ Evaluate$(\phi)$
7        $true\_rewards \leftarrow$ Evaluate$(\mathcal{A})$
8        compare$(true\_rewards, rewards)$
9     **end**
10 **end**
11 **return** $\phi_\mathcal{A}$

12 **Function** evaluate$(\phi\_or\_\mathcal{A})$
13     /* evaluate RL algorithm ($\phi$ or $\mathcal{A}$) on $n$ OOD envs. for the same $T$ timesteps                        */
14     $t \leftarrow 0$
15     $o \leftarrow$ reset$(envs)$
16     $rewards \leftarrow []$
17     **while** $t < T$ **do**
18        $a \leftarrow \phi\_or\_\mathcal{A}(o)$
19        $o, a, r, d \leftarrow$ step$(envs, a)$
20        $\phi\_or\_\mathcal{A} \leftarrow$ append_context$(\phi\_or\_\mathcal{A}, o, a, r, d)$      // only needed for $\phi$
21        $rewards \leftarrow$ append$(rewards, r)$
22     **end**
23     **return** $returns$
24 **end**

---

The important parts of above pseudo-code are marked in red: We need to perform the evaluation of a whole learning process instead of a single episode to evaluate the performance of given AD algorithms.

## A.2 HARDWARE RESOURCES

All the experiments are ran under half-precision `bfloat16` with 3 seeds on a computing server equipped with 12~16 CPUs and a Nvidia A100-80G GPU card.

For parallel data collection, ~2hours are needed for each run.

For training and evaluation of AD algorithms, wall clock time varies from ~2h to ~44h depending on the environment & task settings.

## A.3 ENVIRONMENT SETTINGS

### A.3.1 DETAILS OF EXPERIMENT SETTINGS

We list details of environments and tasks settings in Table S1:

Table S1: Detailed settings of environments & tasks

| settings | train envs | eval envs | grid size | episode length | episodes to collect | total steps collected[α] |
|---|---|---|---|---|---|---|
| dr | | | 9 | 20 | 125 | 2.5e3 |
| dktd | | | 9 | 50 | 150 | 7.5e3 |
| drln | | | 13 | 50 | 150 | 1.25e4 |
| drlq | 1e4 | 1e2 | 13 | 50 | 250 | 8.75e3 |
| drld | | | 15 | 50 | 175 | 1.4e4 |
| dktdln | | | 11 | 70 | 200 | 2e4 |
| dktdlq | | | 9 | 50 | 400 | 2.45e4 |
| dktdld | | | 11 | 70 | 350 | 2.2e4 |

$\alpha$: This is the total environment transitions the algorithm collects and equals to *episode length* $\times$ *episodes to collect*.

## A.4 ALGORITHM DETAILS

### A.4.1 ALGORITHM HYPERPARAMETERS FOR DATA COLLECTION

We use Q-Learning for data collection in all `GridWorld` environments. Its hyperparameters are listed in Table S2:

Table S2: Q-Learning's hyper-parameters

| hyperparameters | values |
|---|---|
| discount | 0.97 |
| learning rate | 1.0 |

### A.4.2 NETWORK STRUCTURE & TRAINING HYPERPARAMETERS

For AD & ET methods, we use the same network structure for a fair comparison. The details of network structure and hyperparameters are listed in Table S3. Their difference mainly locates in the context/chunk length and memory recurrence as detailed in the following sections.

Table S3: Network structure & training hyperparameters

| hyperparameters | values |
|---|---|
| Layers | 4 |
| Hidden dimension | 512 |
| Dropout | 0.1 |
| Feedforward dimension | 2048 |
| Attention head | 64 |
| Optimizer | AdamW |
| Optimizer weight decay | 1e-4 |
| Optimizer betas | (0.9, 0.999) |
| Learning rate | 2e-4 |
| Learning rate scheduler | linear warm-up & cosine decay |
| Batch size | 256 |
| LayerNorm | postnorm |
| Loss function | CrossEntropyLoss |
| Label smooth | 0.1 |

### A.4.3 DETAILS OF ADF TRAINING HYPERPARAMETERS

We list the hyperparameter details of algorithm `ADF` in Table S4.

Table S4: Details of ADF training hyperparameters

| setting | episode length | context length | training steps |
|---|---|---|---|
| dr | 20 | 50 | 1e5 |
| dktd | 50 | 125 | 2e5 |
| drln | 50 | 125 | 2e5 |
| drlq | 50 | 125 | 3e5 |
| drlq | 50 | 125 | 2e5 |
| dktdln | 70 | 125 $^\alpha$ | 4e5 |
| dktdlq | 50 | 125 | 6e5 |
| dktdld | 70 | 125 $^\alpha$ | 6e5 |

### A.4.4  DETAILS OF ADR TRAINING HYPERPARAMETERS

We list the hyperparameter details of algorithm ADR in Table S5.

Table S5: Details of ADR training hyperparameters

| setting | episode length | context length | training steps (ADF $\times$ 5) |
|---|---|---|---|
| dr | 20 | 10 | 5e5 |
| dktd | 50 | | 1e6 |
| drln | 50 | | 1e6 |
| drlq | 50 | | 1.5e6 |
| drld | 50 | 25 | 1e6 |
| dktdln | 70 | | 2e6 |
| dktdlq | 50 | | 3e6 |
| dktdld | 70 | | 2e6 |

### A.4.5  HYPERPARAMETERS OF MEMORY-EFFICIENT ALGORITHMS

We list the hyperparameter details of Memory-Efficient algorithms (MEM, XL and ED) in Table S6.

Table S6: Details of ET training hyperparameters

| settings | chunk length | memory capacity | ADR context length | training steps (ADF $\times$ 5 / ratio) $^\alpha$ |
|---|---|---|---|---|
| dr | 10 | 10 | 10 | 5e5 / 1.5 |
| dktd | | | | 1e6 / 1.7 |
| drln | | | | 1e6 / 1.7 |
| drlq | | | | 1.5e6 / 1.7 |
| drld | 25 | 50 | 25 | 1e6 / 1.7 |
| dktdln | | | | 2e6 / 1.7 |
| dktdlq | | | | 3e6 / 1.7 |
| dktdld | | | | 2e6 / 1.7 |

$\alpha$: we slightly reduced the training steps since the ET methods' effective context is larger than ADR's.

