# OpenReview forum: "Memory-Efficient Algorithm Distillation for In-context Reinforcement Learning"
_ICLR.cc/2025/Conference — Submitted to ICLR 2025_

### Official Review · Reviewer_6qRy · 2024-11-04

**Soundness:** 3
**Presentation:** 3
**Contribution:** 2
**Rating:** 6
**Confidence:** 3

**Summary:**

In this paper, the authors investigate whether efficient Transformer models, designed for long-range document modeling, can be used for Algorithm Distillation (AD) in reinforcement learning (RL). AD generally requires large memory capacity due to the long context of cross-episode state-action-reward histories. Here the authors propose a new benchmark suite to evaluate the in-context learning ability of efficient Transformers in the AD setup. The authors demonstrate that ERNIE-Docs, a variant of Transformer-XL, achieves comparable performance to standard Transformers with significantly reduced memory requirements. They also conduct additional ablation studies analyzing ERNIE-Docs performance for different hyperparameter values.

**Strengths:**

1. The publication is sufficiently clearly written. It clearly states the main motivation, describes the proposed benchmark suite and provides sufficient details about all of the evaluated models.
2. The proposed benchmark could potentially be useful to a wider community once published. And the inclusion of three types of settings (normal, dense and quick) is a valuable component of this benchmark.
3. Overall, the empirical evaluation appears to be well-designed and some of the conclusions (including those related to strong ERNIE-Docs performance and hyperparameter explorations) seem valuable.

**Weaknesses:**

1. While the benchmark itself may hold value, the evaluation of several published and fairly standard algorithms offers limited novelty. The analysis seems to confirm expected performance patterns (including the importance of global positional encodings) and appears to lack deeper insights.
2. The work's current impact is limited by the lack of publicly available code of the proposed benchmark (making it difficult to judge about its potential impact on a broader scientific community).
3. Also, while the GridWorld-based environments may provide a good initial testing ground, further validation across a wider range of environments is very important. Including diverse environment types, as demonstrated in prior work on Algorithm Distillation, would strengthen the generalizability of the findings.

**Questions:**

1. Would it be possible or insightful to interpolate between three proposed settings (normal, dense and quick) to provide a more controllable and continuous way of setting the environment complexity?
2. One of the proposed advantages of the AD method is its data efficiency and the possibility to subsample the source training history. Can there be something said about the behavior of efficient Transformer architectures on sub-sampled episode histories?

---

### Official Review · Reviewer_ZN9X · 2024-11-04

**Soundness:** 2
**Presentation:** 3
**Contribution:** 2
**Rating:** 6
**Confidence:** 3

**Summary:**

The paper investigates the application of Memory-Efficient transformers under the Algorithm Distillation for that leverages in-context learning abilities of autoregressive transformers for better generalization. Specifically, the authors propose a benchmark suite that efficiently assesses ICL capabilities within AD. The benchmark is demonstrated by testing on various memory-efficient transformers. The ERNIE-Docs (ED), a variant of Transformer-XL structure, is highlighted for its competitive performance with reduced memory requirements.

The benchmark suite covers diverse reinforcement learning tasks, including scenarios with sparse and dense rewards, requiring credit assignment and exploration-exploitation balancing. The suite utilizes both JAX for efficient parallel execution and compatibility with PyTorch. Experimental results reveal that ED performs comparably to standard AD but with a lower memory footprint, showing promise for resource-constrained environments. Notably, the study identifies key factors like positional encoding, model size, attention settings, and memory capacity as influential to the ICL performance of ED.

**Strengths:**

+ Novel Application of Efficient Transformers in In-Context Reinforcement Learning: The paper introduces a unique connection by leveraging Efficient Transformer (ET) architectures, specifically ERNIE-Docs, within Algorithm Distillation (AD) for in-context reinforcement learning (ICL).

+ Comprehensive Benchmarking Suite for Memory-Efficient Algorithm Distillation: The paper’s proposed benchmark suite is tailored to evaluate ICL in AD using memory-efficient transformers. The benchmark suite facilitates thorough testing of memory-efficient AD methods across various reinforcement learning challenges, fulfilling the proposed property for meta RL benchmarking.

+ Systematic Ablation Studies and Insights into Hyperparameters: The paper provides detailed ablation studies, shedding light on how positional encoding, model size, attention settings, and recurrence memory capacity impact ICL performance.

**Weaknesses:**

+ Lack of Benchmarking Comparisons: Although the paper’s main contribution is a new benchmarking suite, it does not include comparisons with existing benchmarks. This omission makes it challenging to assess whether the proposed benchmark genuinely outperforms or offers advantages over established alternatives in terms of efficiency, coverage, or flexibility.

+ Limited Range of Tested Methods: As a benchmarking study, the paper evaluates a relatively narrow set of methods, primarily focusing on ERNIE-Docs and Transformer-XL. This limited scope overlooks a wide array of memory-efficient transformers in the literature, reducing the generalizability of the benchmark results and their applicability across diverse models.

+ Engineering Efforts Framed as Research Contributions: While the paper uses JAX for parallelization to achieve computational efficiency, this is more of an engineering choice than a novel research contribution. Clearly distinguishing implementation decisions from research contributions would help emphasize the true innovations in the benchmark, such as findings from ablation studies or unique design features of the benchmarking suite.

**Questions:**

- Could the authors provide comparisons of proposed benchmark with existing benchmarks like Meta-world and DeepMind Control Suite? Discuss the benchmark’s strengths in applicability, memory, runtime, task diversity, and flexibility.

- Could the authors consider testing a broader range of methods? Including additional memory-efficient transformers, such as Longformer, Linformer, and Performer, could help demonstrate the benchmark’s robustness across diverse architectures. Alternatively, the authors should explain in detail why such methods are not comparable for paper`s scope.

- Could the authors make better clarification the distinction between engineering choices and research contributions? For example, framing the use of JAX as an implementation decision for efficiency might help focus attention on unique research insights, such as ablation study findings or design elements specific to in-context learning in memory-constrained environments.

---

> ### Comment · Reviewer_ZN9X · 2024-12-03
> **Response to Author reviews**
>
> I appreciate the detailed responses. My concerns have been addressed. I am increasing my rating from 5 to 6.

---

### Official Review · Reviewer_X3fG · 2024-11-05

**Soundness:** 2
**Presentation:** 2
**Contribution:** 2
**Rating:** 5
**Confidence:** 3

**Summary:**

The paper proposes to use existing recurrent transformers to do Offline RL. Additionally they propose several simple benchmark tasks  to measure quality of their models.

The contribution is a step of simple benchmarks, operating on a small grid. The  tasks include finding a goal point on that grid, and a
more complex DarkKeyToDoor - that requires finding a key first followed by the door. The other part of the benchmark is the reward structure which can vary from the standard sparse reward (e.g. getting to the goal), intermediate sparse -both key and door,
and a dense reward which incorporates the distance to the current target.

**Strengths:**

The paper is well written and for the most part easy to follow. The benchmark dataset is described in great details.

**Weaknesses:**

* The novelty is pretty limited. The benchmark set is pretty straightforward, and it is not entirely obvious how is it different from other similar grid-environments such as mini-grid.

* The best results seem to to be attained at "Dense reward" setting, which is the least challenging, and is generally not very scaleable.

* If i understand correctly the study, the length of the chunk is comparable to the length  of the episode (or 1/2 of the episode, so it is barely even a recurrent network). Since the paper claims memory efficiency, it would benefit greatly  if authors can show that their method can scale to larger ratio between episode length and chunk size.

* Comparison to some standard Grid-World or MiniGrid environments would be helpful (e.g. environments with obstacles)

**Questions:**

1. It is not very clear if you you incorporate  a single or multiple episodes into a single context? I couldn't find any reference in the paper about multiple episodes, but this implies the context length is limited to to 20-70, whereas on figure 11 your single chunk length varies between 20 and 70. Does it mean that that at the upper limit you don't use recurrence at all and the lower limit the recurrence transition is only used at most once?

2. What is "Recurrence capacity" -- is it the number of tokens passed between chunks? Why is it so large? It almost like we don't loose any information when going between chunks - especially considering that your episode length is so small.

---

### Official Review · Reviewer_bBam · 2024-11-09

**Soundness:** 2
**Presentation:** 3
**Contribution:** 2
**Rating:** 5
**Confidence:** 4

**Summary:**

This paper presents an analysis of memory-efficient Algorithm Distillation (AD) in Reinforcement Learning (RL). The authors evaluate the effectiveness of Recurrence Transformers for memory-efficient AD in modified Gridworld environments (DarkRoom and DarkKeyToDoor) with varying grid sizes and reward functions. They further examine how positional encoding and model capacity impact performance.

**Strengths:**

- The study of memory-efficient AD contributes to advancing the practical applicability of framing RL as an autoregressive problem.

**Weaknesses:**

- The behavior of the proposed algorithm is not clearly discussed. In Figure 5, it appears that ED and XL perform worse than other methods on the drld task, despite the claim that dense rewards benefit AD. Additional insights into this result would be beneficial.
- Including plots of performance over time steps could help clarify the behavior of the algorithms.
- In the ablations (Section 5.2), consistency in focusing on the proposed method (ED) would improve clarity. Ablating positional encoding on ED rather than ADF would align better with the primary focus on ED.

**Questions:**

1. What is the intuition behind XL’s difficulty in learning in-context, while MEM—a method that learns global memories—outperforms it?

---

### Meta-Review · Area_Chair_ahti · 2024-12-18

**Metareview:**

The authors investigate whether auto-regressive models achieve a comparable in-context learning ability and can be used for memory-efficient algorithm distillation. In addition, the authors propose a benchmark suite and experiment with ED (ERNIE-Docs) and its hyperparameter tuning.

The reviewers had a mixed assessment of the strengths and weaknesses of the paper. While the paper is well written and the benchmark dataset is described in great detail, the novelty of the work is limited. Furthermore, the experimental setup of the paper is focused on GridWorld-based environments.

Considering the limitation in novelty, the AC recommends a more thorough empirical investigation of the merits of the ED method, along the directions recommended by the reviewers.

I assess that the paper is not yet ready for acceptance and I recommend the authors improve the quality of the experimental setup.

**Additional Comments On Reviewer Discussion:**

The authors and reviewers engaged in productive discussions during the rebuttal. However, the authors had divergent opinions on the adequacy of the experimental setup, in particular concerning the choice of RL environments.

---

### Decision · Program_Chairs · 2025-01-22

Reject